# Genomic Selection in Chinese Holsteins Using Regularized Regression Models for Feature Selection of Whole Genome Sequencing Data

**DOI:** 10.3390/ani12182419

**Published:** 2022-09-14

**Authors:** Shanshan Li, Jian Yu, Huimin Kang, Jianfeng Liu

**Affiliations:** 1Guangdong Provincial Key Laboratory of Animal Molecular Design and Precise Breeding, School of Life Science and Engineering, Foshan University, Foshan 528225, China; 2College of Animal Science and Technology, China Agricultural University, Beijing 100193, China

**Keywords:** Chinese Holstein, whole genome resequencing, milk production trait, genomic selection, feature selection

## Abstract

**Simple Summary:**

Genomic selection (GS) is increasingly widely used in animal breeding, owing to its high efficiency in the genetic improvement of economic traits. In China, GS has been implemented for genetic evaluation of young bulls in dairy cattle breeding programs since 2012. GS is commonly based on single nucleotide polymorphism (SNP) chips. The cost of whole genome sequencing (WGS) has decreased tremendously in recent years, allowing increased studies of WGS-based GS. In this study, based on the imputed WGS data of approximately 8000 Chinese Holsteins, we investigated the performance of GS of milk production traits using the feature selection method of regularized regression. The results showed that WGS-based GS using regularized regression models and the commonly used linear mixed models achieved comparable prediction accuracies. For milk and protein yields, GS using a combination of SNPs selected with a regularized regression model and 50K SNP chip data achieved the best prediction performance, and GS using SNPs selected with a linear mixed model combined with 50K SNP chip data performed best for fat yield. The proposed method of GS based on WGS data, i.e., feature selection using regularization regression models, provides a valuable novel strategy for genomic selection.

**Abstract:**

Genomic selection (GS) is an efficient method to improve genetically economic traits. Feature selection is an important method for GS based on whole-genome sequencing (WGS) data. We investigated the prediction performance of GS of milk production traits using imputed WGS data on 7957 Chinese Holsteins. We used two regularized regression models, least absolute shrinkage and selection operator (LASSO) and elastic net (EN) for feature selection. For comparison, we performed genome-wide association studies based on a linear mixed model (LMM), and the *N* single nucleotide polymorphisms (SNPs) with the lowest *p*-values were selected (LMM_LASSO_ and LMM_EN_), where *N* was the number of non-zero effect SNPs selected by LASSO or EN. GS was conducted using a genomic best linear unbiased prediction (GBLUP) model and several sets of SNPs: (1) selected WGS SNPs; (2) 50K SNP chip data; (3) WGS data; and (4) a combined set of selected WGS SNPs and 50K SNP chip data. The results showed that the prediction accuracies of GS with features selected using LASSO or EN were comparable to those using features selected with LMM_LASSO_ or LMM_EN_. For milk and protein yields, GS using a combination of SNPs selected with LASSO and 50K SNP chip data achieved the best prediction performance, and GS using SNPs selected with LMM_LASSO_ combined with 50K SNP chip data performed best for fat yield. The proposed method, feature selection using regularization regression models, provides a valuable novel strategy for WGS-based GS.

## 1. Introduction

Genomic selection (GS), an upgraded form of marker-assisted selection (MAS), aims to use the genetic effects of genome-wide molecular markers to determine the genomic estimated breeding value (GEBV) of individuals based on optimized statistical models [1]. In GS, the genome is densely covered with markers that are expected to be in complete or partial population-wide linkage disequilibrium (LD) with quantitative trait loci (QTL), allowing a high fraction of genetic variance to be explained by the markers. Generally, GS achieves a higher prediction accuracy than traditional methods based on pedigree. In practice, GS is becoming a popular tool for genetic evaluations of livestock and poultry; it is at present replacing the traditional methodology of genetic evaluation in dairy cattle [2,3], and has been widely used in the commercial breeding of other animals, such as beef cattle [4], pigs [5], chickens [6,7], and sheep [8,9]. In GS using single nucleotide polymorphism (SNP) chips, typical statistical methods include genomic best linear unbiased prediction (GBLUP) and Bayesian methods. In practical applications, GBLUP is favored for its robust predictions and speed.

Decreased costs for whole-genome sequencing (WGS) have provided an exceptional opportunity to systematically detect genetic variants throughout the entire genomes of numerous individuals [10]. In theory, GS based on WGS data can utilize causative mutation sites, and can thus achieve higher accuracy than GS based on SNP chips. This greatly mitigates the negative impacts that the passage of numerous generations between the reference and candidate populations can have on prediction accuracy. In addition, prediction of GEBV based on causative mutation loci can minimize the influence of selection (artificial or natural) on GS accuracy and allow cross-population and cross-breed GS.

Meuwissen and Goddard [11] published the first simulation study of GS based on WGS data, which confirmed that prediction accuracy could be improved by using WGS data. Since then, WGS-based GS has become a topic of great research interest in the fields of livestock [9,12] and poultry breeding [13], and has yielded valuable results. Due to the implementation of the 1000 Bull Genomes Project (http://1000bullgenomes.com/, accessed on 10 August 2022), most relevant research has been undertaken with dairy cattle [12,14,15]. 

SNP chips can typically detect tens or hundreds of thousands of SNPs, whereas WGS data can be used to identify millions of SNPs. The use of such ultra-high-dimensional data causes two key challenges in GS. First, irrelevant and redundant information exists in predictive variables (i.e., SNPs), impacting the prediction performance [16,17,18]; second, computational complexity is greatly increased [19]. Feature selection is an important method in high-dimensional data analysis and modeling. It can improve prediction performance and reduce computational complexity by removing irrelevant and redundant features while retaining relevant features. 

Feature selection has become an important strategy for GS using WGS data. Feature selection of predictive variables is often carried out based on genome annotation information or genome-wide association study (GWAS) results. Studies have shown that feature selection using genomic annotation information (such as gene location) can improve the prediction performance of GS to some extent [20]. There are two main strategies for feature selection based on GWAS results: selection based on the association degree between a SNP and a trait (*p*-value), and selection based on the estimated effect or effect variance of a SNP. Brondum, et al. [21] built a model including SNPs that were found to be significantly associated with traits based on WGS data, and achieved a ~0.005–0.04 times higher reliability compared to SNP chip data. Raymond, et al. [16] used different thresholds of *p*-values to select SNPs from WGS data, and found that, compared with SNP chips, using only the selected SNPs could significantly improve the prediction accuracy, whereas the increase in accuracy was small when both the selected SNPs and SNP chips were used. Ye, et al. [22] found that the prediction accuracy was 1–3% higher when all imputed SNP loci were used rather than SNP chip data. Different thresholds of *p*-values were further used to select loci for prediction, and the accuracy was unchanged compared to the model using all SNPs. Starting with a set of SNP chip data, Warburton, et al. [15] added significantly associated loci based on WGS data, and the top 100 and 250 SNPs were selected based on *p*-values; they found that for the 50K and 800K SNP chips, adding the selected loci resulted in little improvement in accuracy. Among feature selection strategies (SNP chip + SNPs selected based on *p*-value/effect estimation/effect variance), VanRaden, et al. [18] found that methods based on effects and effect variance of SNPs could significantly improve the prediction performance of GS. 

Although the aforementioned feature selection methods have been used to improve the prediction performance of GS to varying degrees, they have several shortcomings. First, the current annotations of livestock and poultry genomes are not sufficiently accurate, so feature selection based on genome annotation information cannot be used to generate models with optimal performance [23]. Second, relevant loci are not selected accurately and comprehensively based on GWAS results; SNP significance values are affected by population size and other factors. Selection of only SNPs significantly associated with a trait may exclude other SNPs that also contribute to the phenotype of interest, resulting in limited improvement in prediction performance. Similarly, arbitrarily selecting a certain number of SNPs based on GWAS results cannot guarantee that all loci in the selected subset will be useful for prediction, or that truly effective loci will not be omitted. A method for accurate, comprehensive selection of relevant features is the key problem in feature selection. 

Regularized regressions are a group of classical methods for variable selection. They identify outcome-associated features, estimate non-zero parameters simultaneously, and are particularly useful for high-dimensional datasets with small sample sizes. Commonly used regularized regression models include least absolute shrinkage and selection operator (LASSO) [24], ridge regression (RR) [25], and elastic net (EN) [26]. RR regularization shrinks predictors and thus makes parameter estimates more stable, whereas LASSO regularization compresses many regression coefficients to zero, thus enabling automatic variable selection (i.e., selecting only one predictor among relevant predictors). EN regularization employs both RR and LASSO penalties, making full use of the advantages of both methods. LASSO and EN automatically perform variable selection. Here, we therefore used LASSO and EN as feature selection methods to perform GS based on WGS data. The purpose of this study was to investigate the application of LASSO and EN models in feature selection to determine whether they could improve the prediction performance of GS models for milk production traits in Chinese Holstein cattle

## 2. Materials and Methods

The cattle used in this experiment came from various farms all over the country. All experimental procedures were approved by the Institutional Animal Care and Use Committee (IACUC) (Beijing, China) at the China Agricultural University (permit number DK996). 

### 2.1. Data

#### 2.1.1. Animals and Phenotype

Estimated breeding values (EBV) of milk yield (MY), fat yield (FY), and protein yield (PY) for 7957 Chinese Holstein cattle were provided by the Dairy Data Center of China. They were estimated based on a random regression model using pedigree and milk production records. De-regressed proofs (DRP) and reliability were calculated using pedigrees and EBV; this was conducted using the method and calculation program proposed by Garrick, et al. [27].

#### 2.1.2. Resequencing Data and Analyses

According to the data requirements of 1000 Bull Genomes Project (Run 8), a total of 86 individuals (76 bulls and 10 cows) with EBV reliability values > 0.8 were selected for re-sequencing. Moreover, this ensured that the sequences of that Chinese Holsteins were appropriately represented in the reference population for imputation. DNA was extracted from semen samples from the bulls and blood samples from cows using the routine procedures. Whole genome sequencing was conducted for each individual. Libraries were prepared using a Twist Library Preparation Enzymatic Fragmentation Kit and sequenced on an Illumina HiSeq X Ten system. The average sequencing depth was 9.84×. First, Trimmomatic v0.38 [28] was used to perform quality control on the FASTQ files. The high-quality paired-end reads in FASTQ format were mapped to the cattle reference genome ARS-UCD1.2 using BWA v0.7.17 [29]. The mapping files in Sequence Alignment/MAP (SAM) format were converted to the binary version of SAM (BAM) format and sorted with SAMtools v1.8 [30]. Next, Picard v2.18.2 (https://broadinstitute.github.io/picard, accessed on 15 May 2020) was used to remove PCR duplicates. Base-quality score recalibration (BQSR) was performed in GATK 3.8 [31]. Finally, BAM and compressed zipped variant call format (g.vcf.gz) files were submitted to the 1000 Bull Genomes Project.

#### 2.1.3. Genotypes

Genotyping was performed for 7957 individuals using the Illumina BovineSNP50 v1 or v2 Beadchip. For the 50K SNP chips, SNP positions were converted to ARS-UCD1.2 genome positions using the UCSC liftover command. Quality control was carried out using PLINK [32]. Individuals and autosomal SNPs that failed to meet one of the following criteria were removed: individual call rate > 0.95; SNP call rate > 0.95; minor allele frequency (MAF) > 0.01; and Hardy–Weinberg Equilibrium test *p*-value > 1.0 × 10^−6^. SNPs on sex chromosomes were excluded from the analysis. After this filtering step, 7957 individuals and 34,384 SNPs remained. 

Genotypes were imputed using Beagle 4.1 [33]; 1059 Holstein cattle from Run8 [34] in the 1000 Bull Genomes Project were used as the reference population and SNPs were called based on the ARS-UCD1.2 reference genome. Quality control was carried out on imputed SNPs using PLINK. Variants were filtered using the following criteria: MAF > 0.01 and Hardy–Weinberg Equilibrium test *p*-value > 1.0 × 10^−6^. After filtering, 7957 individuals and 5,912,383 SNPs remained. SNPs were coded based on the number of copies of the minor allele: 0, 1, and 2 for the minor allele homozygote, the heterozygote, and the major allele homozygote, respectively. 

### 2.2. Statistical Methods

To improve the performance of GS for MY, FY, and PY, SNPs were selected from the WGS data as genomic feature markers. More specifically, they were selected based on GWAS using single-marker mixed linear model and regularized regression models. GBLUP was then used for GS.

#### 2.2.1. SNP Pre-Selection Based on GWAS Results

To link SNPs with complex traits, GWAS was performed separately for MY, FY, and PY using the reference population. The imputed SNPs were pre-selected based on *p*-values. Univariate association tests were performed using a linear mixed model (LMM) implemented in GEMMA v0.98.1 [35], as follows:(1)y=1μ+Xb+Zg+e
where **y** is a vector of DRPs for milk production traits (MY, FY, and PY) of individuals in the reference population; *μ* is the overall mean; **1** is a vector of ones; *b* is a fixed effect of the tested SNP; **X** is a vector of genotypes (0, 1, or 2); **g** is a vector of random polygenic effects with the normal distribution $N(0,G\sigma_{g}^{2})$ (where **G** is the genomic relationship matrix and $\sigma_{g}^{2}$ is the variance of the polygenic effect); **Z** is the corresponding design matrix; and **e** is a vector of residuals, which is assumed to follow a normal distribution of $N(0,I\sigma_{e}^{2})$ (where **I** is the identity matrix and $\sigma_{e}^{2}$ is error variance). 

To prevent possible bias from fitting the same SNP twice, **G** matrices were constructed based on all SNPs except those on the same chromosome as the tested SNP [36]. **G** matrices were calculated as follows [37]:(2)$$G_{ik}=\frac{1}{N}\sum_{i}G_{ijk}=\left\{\begin{array}{l} \frac{1}{N}\sum_{i}\frac{\left(x_{ij}-2p_{i})(x_{ik}-2p_{i}\right)}{2p_{i}(1-p_{i})},j\neq k\\ 1+\frac{1}{N}\sum_{i}\frac{\left(x_{ij}^{2}-(1+2p_{i})x_{ij}+2p_{i}^{2}\right)}{2p_{i}(1-p_{i})},j=k \end{array}\right\}$$
where *x_ij_*(*x_ik_*) is the genotype (coded 0, 1, or 2) for the *i*th SNP of the *j*(*k*)th individual; *N* is the total number of markers; and *p_i_* is the MAF of the *i*th SNP.

The Wald test was used to test the hypothesis for the substitution effect of each SNP. SNPs with *p* < 0.05 for each trait were retained for subsequent analyses.

#### 2.2.2. Feature Selection

For the pre-selected SNPs, feature selection was performed with the regularized regression models LASSO and EN. Both are multi-locus linear regression models, for which the basic linear regression model was:(3)$$y=\mu 1+\sum_{i=1}^{m}X_{i}b_{i}+e$$

The LASSO estimator obtains a sparse solution using *L1* ($\sum_{1}^{m}\left|b_{i}\right|$) penalized least squares:(4)$$\hat{b}(LASSO)=\underset{b}{\mathrm{argmin}}\{(y-\mu 1-Xb)’(y-\mu 1-Xb)+\lambda \sum_{i=1}^{m}\left|b_{i}\right|\}$$
where *λ* is a complexity parameter that controls the amount of shrinkage [26], which is determined from cross-validation.

The EN model uses a mixture of *L1* and *L2* ($\sum_{1}^{m}b_{i}^{2}$) penalties and can be formulated as:(5)$$\hat{b}(EN)=\underset{b}{\mathrm{argmin}}\{(y-\mu 1-Xb)’(y-\mu 1-Xb)+\lambda \sum_{i=1}^{m}\left[(1-\alpha )b_{i}^{2}+\alpha \left|b_{i}\right|\right]\}$$
where *α* is the parameter determining the mix of the *L_1_* and *L_2_* penalties [26], and its value is also obtained via cross-validation.

For each trait, the value of the tuning parameters in LASSO and EN were determined based on the one standard error (1SE) rule [38] via ten-fold cross-validation; in EN, the minimum and maximum values of α were 0.05 and 0.95, respectively, increasing by 0.1. The ‘GLMNET’ package in R was employed for LASSO and EN computation [39]. SNPs with non-zero coefficients in LASSO and EN were selected for subsequent analyses. We also used a feature selection strategy commonly employed in GS studies: for each trait, based on GWAS using LMM, *n_1_* and *n_2_* SNPs with the lowest *p*-values were selected (LMM_LASSO_ and LMM_EN_) for subsequent analyses, where *n_1_* and *n_2_* were the numbers of SNPs selected by LASSO and EN, respectively.

#### 2.2.3. Statistical Models for GS

We compared the performance of GBLUP using four classes of information: (i) only WGS SNPs selected with LASSO, EN, LMM_LASSO_, or LMM_EN_; (ii) only 50K SNP chip data; (iii) both selected SNPs and 50K SNP chip data; and (iv) all imputed SNPs. The GBLUP model was as follows:(6)y=1μ+Za+e
where **y** is a vector of DRPs; **1** is a vector of ones; *μ* is the overall mean; **a** is a vector of breeding values; **Z** is the corresponding incidence matrix; and **e** is a vector of the residuals. It is assumed that $a\sim N(0,G\sigma_{a}^{2})$ and $e\sim N(0,D\sigma_{e}^{2})$. **G** is an additive genomic relationship matrix constructed using the first of the methods described by VanRaden [40]; $\sigma_{a}^{2}$ is additive genetic variance. **D** is a diagonal matrix, $d_{jj}=\left(1-r_{DRP}^{2}\right)/r_{DRP}^{2}$, to account for heterogeneous residual variances ($\sigma_{e}^{2}$) due to different reliabilities of DRP ($r_{DRP}^{2}$).

As stated above, method iii) involved the use of 50K SNP chip data and selected SNPs from WGS data. To take into account the heterogeneity of these two sets of SNPs, we also employed the following GBLUP model for comparison:(7)$$y=1\mu +Z_{50K}a_{50K}+Z_{FS}a_{FS}+e$$
where **a_FS_** and **a_50K_** are vectors of additive genetic effects accounted for by the selected SNPs and 50K chip SNPs (with overlapping SNPs removed from the 50K dataset), respectively; and **Z_FS_** and **Z_50K_** are the corresponding incidence matrices. It is assumed that $a_{\mathrm{FS}}\sim N(0,G_{FS}\sigma_{a_{FS}}^{2})$ and $a_{50K}\sim N(0,G_{50K}\sigma_{a_{50K}}^{2})$. **G_FS_** and **G_50K_** are G matrices constructed from selected SNPs and 50K SNP chip data (with overlapping SNPs removed from the 50K dataset); $\sigma_{a_{FS}}^{2}$ and $\sigma_{a_{50K}}^{2}$ are additive genetic variances explained by SNPs in **G_FS_** and **G_50K_**, respectively. The other terms are defined as detailed above.

The G matrix in the GBLUP model was constructed using gmatrix for Methods i-iii and with the gmatrix module in GMAT v1.0.1 (GitHub - chaoning/GMAT, accessed Jan. 13, 2022) for Method iv. The DMUAI module of DMU v6 release5.2 [41] was used to estimate GEBV.

### 2.3. Evaluation of Prediction Performance

To investigate the prediction performances of different strategies, the 7957 Chinese Holstein cattle were divided into reference and validation populations based on the year of birth. The 6875 individuals born before 2013 were assigned to the reference set, and the 1082 born between 2013 and 2015 were assigned to the validation set. Similarly to our previous study [42], individuals with reliabilities of DRPs lower than 0.40 were removed from the validation set. Prediction accuracy of GS on the validation set was calculated as the correlation between DRP and GEBV, corrected with the root of the average DRP reliability. The regression coefficient of DRP on GEBV was calculated to evaluate prediction bias.

## 3. Results

### 3.1. Feature Selection Based on LASSO and EN

From univariate single marker GWAS analyses, there were 295,045 SNPs for MY, 275,906 for FY, and 294,721 for PY with *p* < 0.05 (Table 1). These SNPs were used to perform feature selection with LASSO and EN. The optimal *α* values of all three milk production traits were 0.05, whereas the *λ* values were diverse (Table 1). The numbers of SNPs determined by LASSO to have non-zero effects were 3546 (MY), 4287 (FY), and 3575 (PY); for EN, there were 11,970 (MY), 12,219 (FY), and 10,497 (PY). 

### 3.2. GS Prediction Performance Using Different Feature Selection Methods

The prediction accuracies of GBLUP using SNPs obtained with different feature selection strategies are presented in Table 2. Overall, GS using SNPs selected from GWAS results (LMM_EN_ or LMM_LASSO_) achieved the highest accuracy for all three milk production traits. Accuracies for models utilizing SNPs selected by EN were 1.65%, 4.21%, and 15.65% higher (for MY, FY, and PY, respectively) compared to the accuracies of the corresponding models utilizing SNPs selected by LASSO. 

For MY, there were very small differences in the accuracy of GS when feature selection was conducted with EN compared to LMM_EN_ or with LASSO compared to LMM_LASSO_. Specifically, GS using LMM_EN_ for feature selection was 1.33% more accurate than when EN was used, whereas GS using LMM_LASSO_ was 1.50% less accurate than when LASSO was used. For FY, prediction accuracy was 3.38% higher for GS using EN compared to LMM_EN_. However, the prediction accuracy was nearly 20% less for GS when LASSO was used compared to LMM_LASSO_. For PY, the accuracy of GS using LMM_LASSO_ was 18.95% higher than with LASSO, and the accuracy of GS using LMM_EN_ was 8.37% higher than with EN.

The regression coefficients of DRP on GEBV differed between feature selection methods, reflecting prediction biases (Table 3). An optimal prediction would have a regression coefficient of 1. GS using feature selection based on GWAS (LMM_EN_ and LMM_LASSO_) showed less bias than direct feature selection with EN or LASSO. For all traits, GS using SNPs selected with LMM_LASSO_ had the least bias (range = 0.566–0.685). Consistent with the higher accuracy, GS using EN feature selection resulted in less biased predictions (0.200–0.267) than when LASSO was used (0.183–0.248). 

### 3.3. Prediction Performance of GS Using Both 50K SNP Chip Data and Selected WGS SNPs

GS was performed using a combination of 50K SNP chip data and selected SNPs from WGS; furthermore, two GBLUP models were used, as shown in equations [6,7]. GS was also performed with only 50K SNP chip data and with only WGS SNP data for comparison. 

For the combined datasets of 50K SNP chip data and selected SNPs, the numbers of overlapped SNPs ranged from 22 (50K + LMM_LASSO_) to 114 (50K + EN) for the three traits. The numbers of SNPs used in different scenarios are shown in Table 4. With the datasets of 50K + EN and 50K + LMM_EN_, about 46,000 SNPs were used in GS of MY and FY; and about 44,800 SNPs in GS of PY. With the datasets of 50K + LASSO and 50K + LMM_LASSO_, about 37,900 SNPs were used in GS of MY and PY, and about 38,600 SNPs in GS of FY. 

The prediction accuracy and bias (regression coefficients of DRP on GEBV) of these models are shown in Table 5 and Table 6, respectively. The prediction accuracy of model (6) was higher than that of model (7) for all three milk production traits (Table 5). The GS models using 50K SNP chip data plus WGS SNPs selected with EN (50K+EN) and using 50K SNP chip data plus WGS SNPs selected with LASSO (50K+LASSO) failed to converge, even with 1000 iterations of the model built using equation (7) (model (7)); the accuracies of these models were therefore not taken into account. For PY, the accuracy of GS using 50K+LASSO was nearly twice as high when the model built using equation [6] (model (6)) was used compared to model (7) (0.2903 versus 0.1665). For FY, the accuracy of GS was nearly 30% higher for both 50K+LMM_EN_ and 50K+LASSO when model (6) was used compared to model (7). For all three traits, GS using model (6) had higher accuracy with 50K+LASSO or 50K+LMM_LASSO_ compared to 50K+EN or 50K+LMM_EN_.

For MY, GS using 50K+LASSO had the highest accuracy, which was close to that only using 50K SNP chip or WGS data. Similarly, accuracy was highest for PY with 50K+LASSO (0.2903), although models utilizing 50K SNP chip or WGS data had <3% higher accuracy. For FY, GS using 50K+LMM_EN_ or 50K+LMM_LASSO_ had higher accuracy than all other models; GS using 50K+LMM_LASSO_ had the highest accuracy (0.2905).

Regression coefficients of DRP on GEBV for GS using 50K SNP chip data combined with selected SNPs differed between feature selection methods, showing a range of prediction biases (Table 6). For all traits, model (6) was less biased than model (7). For GS of MY based on model (6), 50K+LMM_EN_ was the least biased, followed by 50K+LASSO. Similarly, GS using 50K+LMM_EN_ was the least biased for PY, followed by 50K+LASSO. For FY, GS using 50K+LMM_LASSO_ and model (6) achieved not only the highest accuracy but also the lowest bias.

## 4. Discussion

The objective of this study was to explore the application of regularized regression feature selection strategies to WGS-based GS for milk production traits in Chinese Holstein cattle. It was previously shown that it may be necessary to improve the prediction accuracy of GS based on WGS data by increasing the number of individuals in the reference population [43] or by selecting feature markers using different strategies [44,45]. The regularized regression models LASSO and EN had not previously been used as feature selection strategies in WGS-based GS [12,45]. LASSO and EN shrink the regression coefficients of minor effect SNPs towards zero, removing irrelevant and redundant features to yield the final set of selected features.

In the present study, GS using 50K SNP chip data and WGS SNPs selected with LASSO (50K+LASSO) had the best prediction performance for MY and PY; the 50K+LMM_LASSO_ method performed best for GS of FY. GS using SNPs selected by EN and LASSO did not lead to higher accuracy than GS using SNPs selected based on *p*-values from single-marker linear mixed model GWAS analysis (LMM_LASSO_ and LMM_EN_), and the bias was much higher in the former. There are four possible reasons for this result; these are enumerated below.

First, there were a total of 5,912,383 SNPs in the WGS data, meaning it was an ultra-high-dimensional dataset. Therefore, the number of SNPs (*p*) was much larger than the number of observations (*n*). Theoretically, LASSO and EN analyses are suitable for such instances where *p* >> *n*. However, the ‘GLMNET’ package in R could not handle such a large volume of data, preventing optimal feature selection. To solve this problem, single-marker GWAS analysis was used for SNP pre-selection prior to the use of LASSO or EN for feature selection. With the single marker model, adjacent SNPs may be selected for subsequent analyses, although they may have low *p*-values due to being in LD with the same causal mutation. This would prevent EN and LASSO from fully exploiting their advantages.

Moreover, it has been shown that performing GS with markers selected based on GWAS results did not significantly improve the prediction performance; when markers with *p*-values < 0.05 were used for prediction, the prediction accuracy of some traits significantly decreased [46]. This indicates that SNP pre-selection based on GWAS may have a negative effect on GS accuracy. We therefore recommend using other pre-selection methods followed by LASSO or EN. 

Ideally, all SNPs in WGS data would be directly analyzed using EN or LASSO. We used the ‘BIGLASSO’ package in R, which can improve the memory and computational efficiency of LASSO models built with ultra-high-dimensional data [47]. Unfortunately, this package was only able to analyze 1,000,000 SNPs. It is therefore necessary to reevaluate feature selection in GS when software becomes available that can perform LASSO or EN with millions or even tens of millions of SNPs. 

Second, we pre-selected SNPs with *p*-values < 0.05 from single-marker GWAS for subsequent analyses. It is possible that the accuracy of GS using LASSO or EN, particularly the latter, was not greatly improved due to the threshold value. Ye, et al. [45] divided SNPs into different categories based on several *p*-value thresholds: 0.05, 0.001, 0.0001, 0.00001, and 0.000001. The results showed that GS using SNPs with different *p*-value thresholds has lower accuracy and larger bias compared to models built with the complete WGS data when using either GBLUP or GFBLUP.

A third possible reason for the undesirable GS results after using LASSO or EN for feature selection is that the statistical model for GS used in this study was GBLUP. Although it performs robustly under a variety of conditions, Meuwissen and Goddard [11] demonstrated that GBLUP does not take full advantage of WGS data. GS based on WGS data tends to have better results when using Bayes series models. In theory, Bayesian methods can utilize all of the mutation information provided by WGS data. van Binsbergen, et al. [2] performed GS using GBLUP and Bayesian stochastic search variable selection (BSSVS) based on WGS data for somatic cell scores, intervals between first and last insemination, and protein yield in Holstein cattle. BSSVS performed better than GBLUP in all cases. Liu, et al. [48] used GBLUP and Bayesian four-distribution mixture models to perform GS for milk production and reproduction traits by integrating additional SNPs selected from imputed WGS data. The Bayesian four-distribution model achieved higher accuracy than the GBLUP model for milk and protein yields.

Finally, selected SNPs were integrated with 50K SNP chip data and regarded as random effects in this study; however, it has been demonstrated that including some selected SNPs as fixed effects or giving them a higher prior may improve prediction accuracy [2]. Therefore, in addition to our approach of constructing two genomic relationship matrices with SNPs selected for traits as random effects in the 50K SNP chip dataset, GS could be attempted with large-effect SNPs selected for traits as fixed effects. Brondum, et al. [21] reported that accuracy may be increased by adding several genomic SNPs found in WGS data to conventional 54K SNP chip data, especially fertility trait data from significant QTLs. This strategy should be tested in future studies.

In our study, the GBLUP model that considered one G matrix (model (6)) achieved higher accuracy than the model that considered two G matrices (model (7)). Brondum, et al. [21] reported that model (6) had higher accuracies than model (7) for all the traits analyzed. In the study by Liu, et al. [48], model (6) performed better than model (7) in GS of mastitis and fertility, but model (7) performed better in GS of milk and protein. Moreover, in GS of fat, model (6) had better performance when using the reference populations of Danish bulls and Danish and US bulls; model (7) had better performance when using the reference population of Danish cows; models (6) and (7) had the same performance when using the reference populations of Danish bulls and cows, as well as Danish and US bulls and Danish cows. Gebreyesus, et al. [49] reported that model (6) achieved better performance in a scenario involving additional SNPs selected by GWAS in Denmark–Finland–Sweden dairy cattle populations; but model (7) achieved higher accuracy in a scenario involving SNPs selected from GWAS for survival index. Therefore, the performance of models (6) and (7) varies depending on the traits, the reference population, and how many SNPs are selected. Moreover, the problems of noise and confounding between the effects may be present in model (7).

## 5. Conclusions

In this study, we compared the performance of different feature selection strategies in GS based on WGS data. The results showed that GS accuracy was slightly lower when SNPs selected with LASSO or EN were used compared to SNPs selected with LMM_LASSO_ or LMM_EN_. In addition to 50K SNP chip data, selected WGS SNPs were integrated to construct one or two G matrices in a GBLUP model. The model incorporating one G matrix achieved better prediction performance. For milk yield and protein yield, GS using the combination of SNPs from the 50K SNP chip and WGS SNPs selected with LASSO achieved the best prediction performance; for fat yield, GS using SNPs from the 50K SNP chip and those selected with LMM_LASSO_ had the best performance.

## Figures and Tables

**Table 1 animals-12-02419-t001:** Optimal parameter values and results of feature selection with LASSO and EN.

Trait	N ^†^	LASSO		EN		
		*λ* ^‡^	*n* ^§^	*α* ^‡^	*λ* ^‡^	*n* ^§^
MY	295,045	0.0044	3546	0.05	0.0730	11,970
FY	275,906	0.0033	4287	0.05	0.0573	12,219
PY	294,721	0.0041	3575	0.05	0.0742	10,497

MY, milk yield; FY, fat yield; PY, protein yield; ^†^
*N*, number of SNPs used in LASSO and EN analyses; ^‡^
*λ* and *α*, optimal parameter values; ^§^
*n*, number of SNPs with non-zero effects.

**Table 2 animals-12-02419-t002:** Prediction accuracy of GS with different feature selection methods.

Trait	EN	LASSO	LMM_EN_	LMM_LASSO_
MY	0.2029	0.1996	**0.2056**	0.1966
FY	0.1807	0.1734	0.1748	**0.2147**
PY	0.1921	0.1661	**0.2285**	0.1800

LMM_EN_ and LMM_LASSO_ were used to select *N* SNPs with the lowest *p*-values from GWAS based on a single-marker linear mixed model, where *N* was the number of SNPs with non-zero effects based on results from EN and LASSO. The highest accuracy of GS for each trait was in bold.

**Table 3 animals-12-02419-t003:** Validation regression coefficients of GS with different feature selection methods.

Trait	EN	LASSO	LMM_EN_	LMM_LASSO_
MY	0.267	0.248	0.538	0.630
FY	0.200	0.183	0.364	0.685
PY	0.249	0.203	0.591	0.566

LMM_EN_ and LMM_LASSO_ were used to select *N* SNPs with the lowest *p*-values from GWAS based on a single-marker linear mixed model, where *N* was the number of SNPs with non-zero effects based on results from EN and LASSO.

**Table 4 animals-12-02419-t004:** The number of SNPs used in different scenarios.

Trait	50K+EN	50K+LASSO	50K+LMM_EN_	50K+LMM_LASSO_	50K	WGS
MY	46,258	37,903	46,280	37,908	34,384	5,912,383
FY	46,489	38,628	46,499	38,637	34,384	5,912,383
PY	44,801	37,930	44,820	37,935	34,384	5,912,383

50K+EN, 50K+LASSO, 50K+LMM_LASSO_, and 50K+LMM_EN_ were genomic selection models combining 50K SNP chip and SNPs selected with different feature selection methods from imputed whole-genome sequence (WGS) data; 50K and WGS were genomic selection models using only 50K SNP chip and only WGS data, respectively.

**Table 5 animals-12-02419-t005:** Prediction accuracy of GS using 50K SNP chip data combined with selected SNPs from WGS data.

Trait	Model	50K+EN	50K+LASSO	50K+LMM_EN_	50K+LMM_LASSO_	50K	WGS
MY	[6]	0.2370	**0.2947**	0.2559	0.2643	0.2962	0.2990
	[7]	-	-	0.2145	0.2237		
FY	[6]	0.1991	0.2222	0.2721	**0.2905**	0.2691	0.2632
	[7]	0.1807	0.1735	0.2069	0.2815		
PY	[6]	0.2434	**0.2903**	0.2589	0.2720	0.2985	0.2975
	[7]	0.1921	0.1665	0.2290	0.1964		

Models (6) and (7) were GBLUP models that considered one or two G matrices, respectively; 50K+EN, 50K+LASSO, 50K+LMM_LASSO_, and 50K+LMM_EN_ were genomic selection models combining 50K SNP chip and SNPs selected with different feature selection methods from imputed whole-genome sequence (WGS) data; 50K and WGS were genomic selection models using only 50K SNP chip and only WGS data, respectively. The highest accuracy of GS using the combination of 50K SNP chip data and selected SNPs for each traitwas in bold.

**Table 6 animals-12-02419-t006:** Validation regression coefficients from GS using 50K SNP chip data combined with selected SNPs from WGS data.

Trait	Model	50K+EN	50K+LASSO	50K+LMM_EN_	50K+LMM_LASSO_	50K	WGS
MY	[6]	0.444	1.117	0.988	1.295	1.826	1.820
	[7]	-	-	0.551	0.675		
FY	[6]	0.258	0.419	0.747	0.977	1.126	1.114
	[7]	0.200	0.182	0.423	0.831		
PY	[6]	0.439	1.049	1.004	1.251	1.702	1.673
	[7]	0.249	0.202	0.584	0.574		

Models (6) and (7) were GBLUP models that considered one or two G matrices, respectively; 50K+EN, 50K+LASSO, 50K+LMM_LASSO_, and 50K+LMM_EN_ were genomic selection models combining 50K SNP chip and SNPs selected with different feature selection methods from imputed whole-genome sequence (WGS) data; 50K and WGS were genomic selection models using only 50K SNP chip and only WGS data, respectively.

## Data Availability

The data presented in this study are available on request. These data are not publicly available to preserve the data privacy of the commercial farm.

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
