# Peer review of "Genomic Selection in Chinese Holsteins Using Regularized Regression Models for Feature Selection of Whole Genome Sequencing Data"

_animals, 2022, doi:10.3390/ani12182419_

Round 1

Reviewer 1 Report

Dear authors,

With the cheapness of WGS , WGS data is more and more used in GS. How to effiecntly select SNP from WGS is the hot topic because all SNP from WGS data do not improve GS accuracy compared with 54k or HD chip, even decreases the accuracy. For this manuscript, it is a good try to use the LASSO and EN models to select SNP from WGS data, then to investigate whether it can improve GS accuracy. I have some comments, and hope that these comments can help improve this manuscript.

1.       Line 70-73  Some related references should be cited.

2.       Line 91-92  there shows too many authors. This is an improper cite. Please modify it and check it throught the manuscript.

3.       Line 174 : please clarify how to measure imputation accuracy. Why use 0.35 as the cutoff?

4.       Line 221-224: For scenarios of LASSO and EN, the SNPs firstly are selected by GWAS, then selected by LASSO or Len. For scenarios of LMMLASSO and LMMEN, the SNPs firstly are selected by LASSO or Len, then selected by GWAS? Please calrify this sentence.

5.       Line 243:  How many SNP are overlapped between selected SNPs and 50k chip SNPs?  Please clarify it in Results part. Why overlapping SNPs removed from the 50k dataset? I think it is better to remove overlapping SNPs from selected SNPs dataset while keeping 50k unchanged.

6.       Line 257 Why choose 0.35 as the threshold to select validation individuals.

7.       Line 258 I think that it is not the correlation between DRP and GEBV. It should be the square of the correaltion while the tables showed the GS accuracy.

8.       Table 4 and 5 It is better to show how many SNPs in each scenario.

9.       Some studies reported that model [7] got higher prediction accuracy than model[6]. But you got different results. You should discussed this point in Discussion part. Some references should be cited to support your results or give some reasons to explain it.      

Reviewer 2 Report

Genomic selection in Chinese Holsteins using regularized regression models for feature selection of whole genome sequencing data by Shanshan Li et al.

This manuscript deals with a “hot topic” of inclusion of sequence data in the genomic selection. It is superbly written and very well organized. I would suggest some improvement in terms of adding more details regarding methods and results. Specific comments are below:

Line 146-160: Please describe the goal of sequencing 86 individuals and submitting the results to the 1000 Bull Genomes Project. Was that done to ensure that the sequences of that specific population are appropriately represented in the reference population?

Line 188: It may be clear, but I would appreciate describing why the Zg term was in the model (to account for the polygenic background).

Line 212: What are L1 and L2 penalties (please define).

Line 213: What are the parameters lambda and alpha (please describe or mention reference).

Line 219-224: Could you please describe in more details the difference between LASSO and LMMLASSO, and EN and LMMEN? If you performed GWAS using SNPs selected by LASSO/EN and then only chose those with the smallest p-values, it would be good to show some results (what was the threshold p-value to include a SNP, etc.)

Line 239: It appears that using 2 G matrices, one for each set of SNPs, contributes to the noise and confounding between the effects. Have you tried putting all the SNPs together (50K + selected with LASSO/EN)?

Line 351 and Table 5: What exactly are the WGS data? Please describe.

Line 91-92; 103-104; 403-404: The reference is cited with all authors instead of first author et al.
